# Why the Coriolis force turns a wind farm wake clockwise in the Northern Hemisphere

Maarten Paul van der Laan[1] and Niels Nørmark Sørensen[1]

[1]Technical University of Denmark, DTU Wind Energy, Risø Campus, Frederiksborgvej 399, 4000 Roskilde, Denmark

*Correspondence to:* Maarten Paul van der Laan (plaa@dtu.dk)

**Abstract.** The interaction between the Coriolis force and a wind farm wake is investigated by Reynolds-Averaged Navier-Stokes simulations, using two different wind farm representations: a high roughness and alternatively by 5×5 actuator disks. Surprisingly, the calculated wind farm wake deflection is opposite in the two simulations. A momentum balance in the cross flow direction shows that the interaction between the Coriolis force and the 5×5 actuator disks is complex due to turbulent mixing of veered momentum from above into the wind farm, which is not observed for the interaction between the Coriolis force and a roughness change. When the wind farm simulations are performed with a horizontally constant Coriolis force in order to isolate the effect of the wind veer, the wind farm wake deflection of the 5×5 actuator disks simulation remains unchanged. This proofs that the present wind veer is deflecting the wind farm wake and not the local changes in the Coriolis force in the wake deficit region. An additional simulation of a single actuator disk, operating in a shallow atmospheric boundary layer, confirms that the Coriolis force indirectly turns a wind turbine wake clockwise, as observed from above, due to the presence of a strong wind veer.

## 1 Introduction

In recent years, wind farms have grown in size and are more frequently placed in wind farm clusters. This means that large scale effects are becoming more important for wind turbine wake interaction in wind farms, and especially for the interaction between wind farms. One large scale effect that is often neglected by wind farm modelers is the effect of the Coriolis force on wind turbine/farm wakes. In previous work (van der Laan et al., 2015a), we have shown that the Coriolis force should not be neglected in Reynolds-Averaged Navier-Stokes simulations of a wind farm cluster consisting of two wind farms in a neutrally stratified atmospheric boundary layer (ABL). The deflection of the upstream wind farm wake resulted in a lower power production of the downstream wind farm, because the Coriolis force aligned the upstream wind farm wake towards the curved wind turbine rows of the downstream wind farm. Note that a constant latitude was used, which means that the global turning of the Coriolis force was not modeled. In other words, only the interaction between the Coriolis force and local disturbances in the velocity field were investigated. In the present work, we will also use a constant latitude.

The literature does not agree on the turning direction of wind farm wakes caused by the Coriolis force. Volker et al. (2015) showed that mesoscale models with different wind farm parameterizations can show wind farm wake deflections in opposite directions for the same test case. Magnusson and Smedman (1993) have observed that a strong wind veer, caused by the

Coriolis force and more pronounced in stable atmospheric conditions, can lead to a skewed wind turbine wake. Several authors have used Large Eddy Simulation (LES) to investigate this effect on wind turbines and wind farms. LES of Lu and Porté-Agel (2011) confirmed that a strong wind veer can lead to a skewed wind turbine wake, which is observed as a clockwise rotation at hub height, in the Northern Hemisphere. Here, we define the wake deflection as clockwise and anticlockwise, as observed

from above and we only discuss wind farms located at the Northern Hemisphere for simplicity. More recently, Churchfield et al. (2016) and Abkar and Porté-Agel (2016) have shown similar results using LES of wakes in a stable ABL. On the contrary, Dörenkämper et al. (2015) reported a small anticlockwise deflection in a LES of a wind farm wake in a stable ABL. Dörenkämper et al. (2015) generated the ABL with a precursor simulation using a different roughness length compared to the one applied in the wind farm simulation in order to model coast effects on an offshore wind farm. This means that the inlet

ABL profile is developing downstream and could have impacted the wind farm wake deflection. Allaerts and Meyers (2017) also observed a small anticlockwise wind farm deflection ($2°$) in LES of a wind farm in a neutral ABL. However, the observed turning is so small that it becomes challenging to extract it from a LES data set. Even though Allaerts and Meyers (2017) have used a precursor simulation with the same the roughness length as used in the wind farm simulation, the inserted ABL at the inlet can still develop downstream, which can lead to small wind direction changes in the wind farm. This is because the

generated ABL from the precursor simulation in LES is only in a pseudo steady state. In our RANS simulations (van der Laan et al., 2015a), the neutral ABL is steady state and it is in balance with the entire domain. Another problem of simulating the interaction between the Coriolis force and a wind farm wake in LES is that one might need to simulate a very long time in order to obtain a statistically independent average of a small quantify such as the wind direction deflection in neutral atmospheric conditions.

Mitraszewski (2012) argued that a wind farm can be seen as a roughness change, and therefore the Coriolis force should turn the wind farm wake anticlockwise, following Orr et al. (2005). In contradiction to Mitraszewski (2012), it was shown in previous work (van der Laan et al., 2015a) that the Coriolis force turns a wind farm wake clockwise (in neutral atmospheric conditions), and explained it as a result of a stream-wise decreasing Coriolis force in the wake recovery region. In summary, the interaction between the Coriolis force and the wind farm wake is explained in the literature in two different ways:

1. The Coriolis forces change in the wake region and induce a clockwise (van der Laan et al., 2015a) or anticlockwise (Mitraszewski, 2012; Dörenkämper et al., 2015; Allaerts and Meyers, 2017) wake defection.

   2. The present wind veer deflects the wind farm wake clockwise (Magnusson and Smedman, 1993; Lu and Porté-Agel, 2011; Churchfield et al., 2016; Abkar and Porté-Agel, 2016).

Our goal is to clarify why the Coriolis force turns a wind farm wake clockwise in the Northern Hemisphere. First, we test the

hypothesis of Mitraszewski (2012) by performing two wind farm simulations with RANS in a neutrally stratified ABL, where the wind farm is represented in two different ways; using actuator disks (ADs) (Mikkelsen, 2003) and using a high roughness in the wind farm area. Secondly, the two wind farm simulations are repeated using a Coriolis force that is set constant in horizontal directions in order to remove the influence of local changes in the Coriolis force in the wake region, and to isolate the effect of the present wind weer on the wind farm wake deflection. In addition, we investigate the wake deflection of a

single AD placed in a shallow ABL, where the wind veer over the rotor area is large to see if our RANS model agrees with LES results from Lu and Porté-Agel (2011); Churchfield et al. (2016); Abkar and Porté-Agel (2016) and field measurements of Magnusson and Smedman (1993).

Note that this work is an extension of van der Laan and Sørensen (2016), presented at the TORQUE 2016 conference.

## 2 Methodology

In order to understand the interaction between the Coriolis force and a wind farm wake, two RANS simulations of a simple rectangular wind farm of 5×5 NREL-5MW wind turbines (Jonkman et al., 2009) are carried out, where the wind farm is represented in two different ways:

1. 25 wind turbines represented by ADs (Mikkelsen, 2003) with variable forces (van der Laan et al., 2015b), without wake rotation.

2. A high roughness of 1 m in the wind farm area.

The NREL-5MW wind turbine has a hub height ($z_H$) and a rotor diameter ($D$) of 90 m and 126 m, respectively. The wind turbines spacing is set to $8D$ in both horizontal directions. In addition, a third RANS simulation of a single NREL-5MW wind turbine in a shallow ABL is carried out, where the wind turbine is represented by an AD.

In the simulation where the wind farm is represented by a high roughness, a roughness length of 1 m is chosen. Frandsen et al. (2009) and Volker (2014) have used values between 0.5-0.7 m, using the wind farm roughness length relation of Frandsen (2007). Calaf et al. (2010) performed LES of a fully developed boundary layer over an infinitely large wind farm and showed that the wind farm roughness length can be as high as 8.9 m. Hence, using 1 m as a wind farm roughness length in the present work seems to be in the correct range. Choosing a different roughness length will change the results, but not the general trends in wind farm wake deflection.

The numerical setup of the RANS simulations with ADs including the Coriolis force is fully described in previous work (van der Laan et al., 2015a), and we will briefly summarize it here. The simulations are carried out with the in-house flow solver EllipSys3D founded by Sørensen (1994); Michelsen (1992). The turbulence is modeled with a modified $k$-$\varepsilon$ model that limits the boundary layer height through a global length scale limiter as introduced by Apsley and Castro (1997), and it includes a local length scale limiter that is necessary to resolve the near wind turbine wake properly (van der Laan et al., 2015c). The inflow profiles of the wind farm simulations are determined from a neutrally stratified precursor simulation, where the Coriolis force is balanced by a defined pressure gradient, both terms are implemented as a momentum source term $S_v$:

$$S_{v,x} = \rho f_c (V - V_G), \qquad S_{v,y} = -\rho f_c (U - U_G), \tag{1}$$

with $\rho$ as the air density, $f_c$ as the Coriolis parameter set to $10^{-4}$ 1/s, $U$ and $V$ are the stream-wise and lateral velocity components, and the subscript $G$ denotes the geostrophic wind, which is set to 12 m/s. A uniform roughness length of $10^{-4}$ m is chosen. In the wind farm simulations, the maximum turbulent length scale $\ell_{t,\max}$ used in the global length scale limiter is

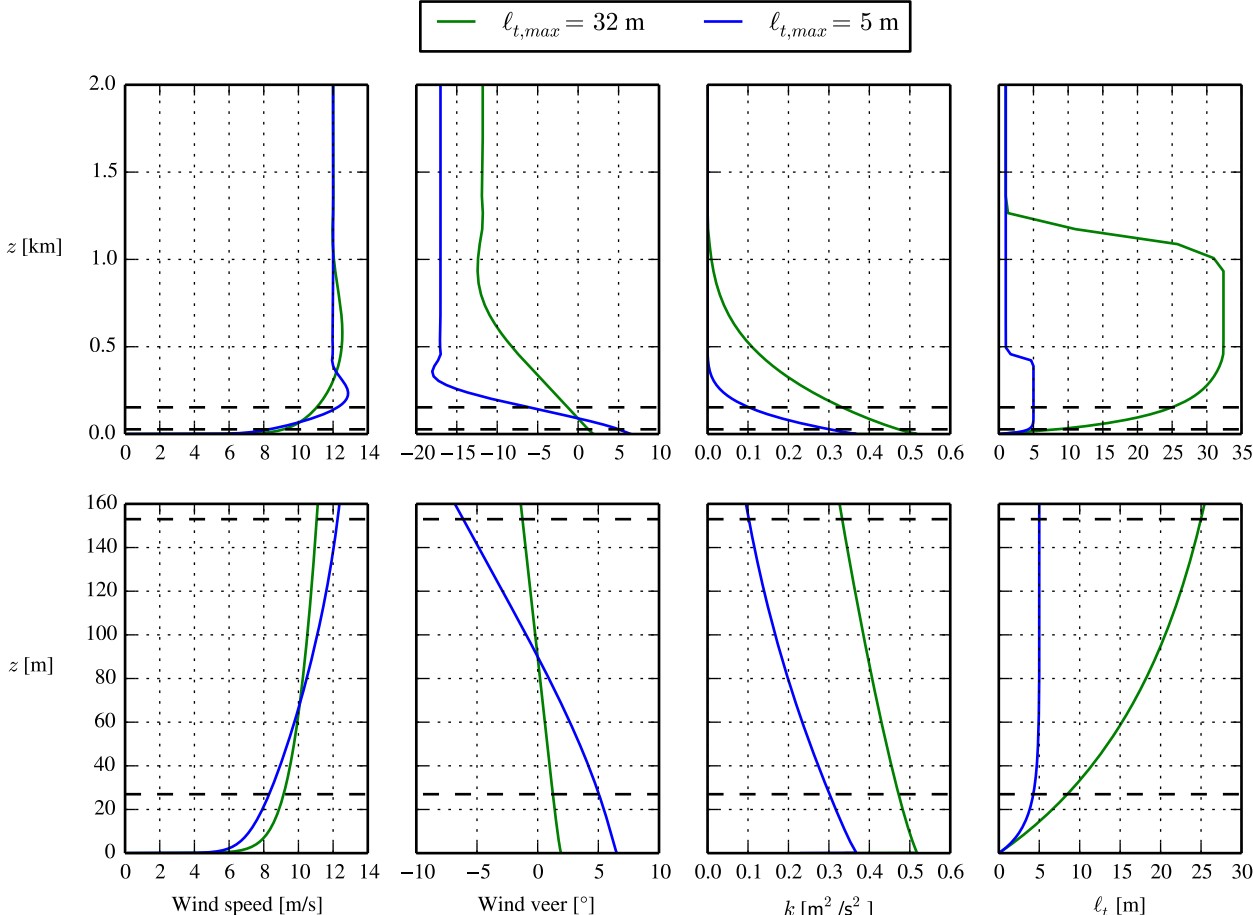

**Figure 1.** Rotated inlet profiles calculated by the precursor. Bottom plots are a zoomed view of the top plots. Rotor area is shown as black dashed lines.

based on Blackadar (1962):

$$\ell_{t,\mathrm{max}} = 0.00027 \frac{G}{f_c}, \tag{2}$$

which gives an $\ell_{t,\mathrm{max}}$ of 32.4 m. In the single AD simulation, $\ell_{t,\mathrm{max}}$ is set to 5 m to enforce a shallow ABL with a strong wind veer over the rotor area. The shallow ABL could be seen as a pseudo stable ABL, where the direct modeling of buoyancy in

5    the momentum and turbulence equations, as performed in the RANS setup of Sogachev et al. (2012); Koblitz et al. (2015), is neglected. The precursor simulation of the wind farm test cases ($\ell_{t,\mathrm{max}} = 32.4$ m) calculates a velocity of 10.4 m/s and a turbulence intensity of 5% at hub height, which represents neutral off-shore conditions. The velocity and a turbulence intensity at hub height for the precursor simulation of the single AD in a shallow ABL ($\ell_{t,\mathrm{max}} = 5$ m) is equal to 10.8 m/s and 3.2%, respectively. The calculated profiles include wind veer, and are rotated to enforce a chosen row aligned wind direction of 270°

10    at hub height in the wind farm simulations. The same wind direction is used in the single AD case. The rotated inflow profiles

from both precursor simulations are shown in Figure 1. A wind veer of about 2.5° and 11°are present in the rotor area, as shown in one of the bottom plots of Figure 1.

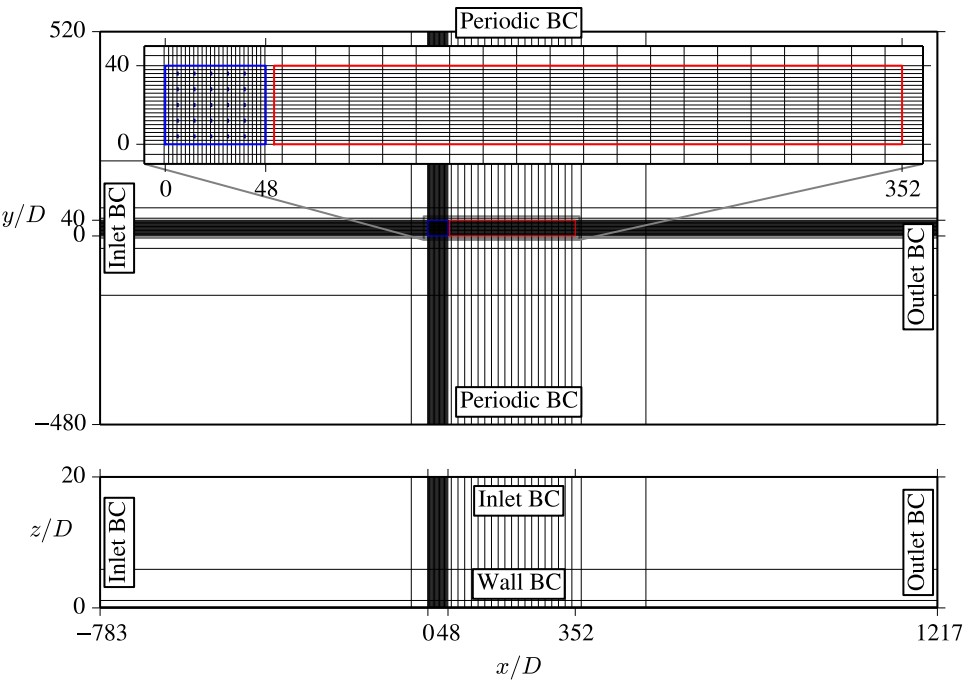

**Figure 2.** Grid and boundary conditions of wind farm simulations. Every 16th grid line is plotted. Top: top view of the grid, including a zoomed view. Bottom: side view of the grid. Blue filled boxes represent ADs. Spacing inside the blue and red rectangles are set to $D/8$ and $D$, respectively.

The same numerical grid is used in both wind farm simulations. The domain definition including wind farm layout and boundary conditions (BC) is shown in Figure 2. The grid is Cartesian and represents a box-shaped domain with dimensions $2000D \times 1000D \times 20D$, in stream-wise, lateral and vertical directions, respectively. The grid consist of 22 million cells, where the horizontal spacing in and around the wind farm (inside the blue rectangle of Figure 2) is $D/8$. Inside the red rectangle of Figure 2, the grid cells are stretched in the stream-wise direction towards a spacing of $1D$, up to $352D \approx 44$ km downstream. The vertical resolution starts with a cell height of 0.5 m and it grows with height using an expansion ratio of about 1.1. The profiles from the precursor simulation are inserted at the inlet BC, as shown in Figure 2. In addition, the top boundary of the domain is also an inlet BC. The lateral boundaries are periodic to account for wind veer. At the outlet BC, a fully developed flow is assumed. A rough wall BC (Sørensen et al., 2007) is placed at the bottom of the domain.

The numerical grid and boundary conditions of the single AD is shown Figure 3. The domain has dimensions $25D \times 16D \times 16D$, in stream-wise, lateral and vertical directions, respectively. The horizontal grid spacing inside the blue rectangle with dimensions $12D \times 3D$ is set to $D/8$. A total number of $7.9 \times 10^5$ cells are used in the grid. The boundary conditions of the wind farm simulations are also applied to the single AD simulation.

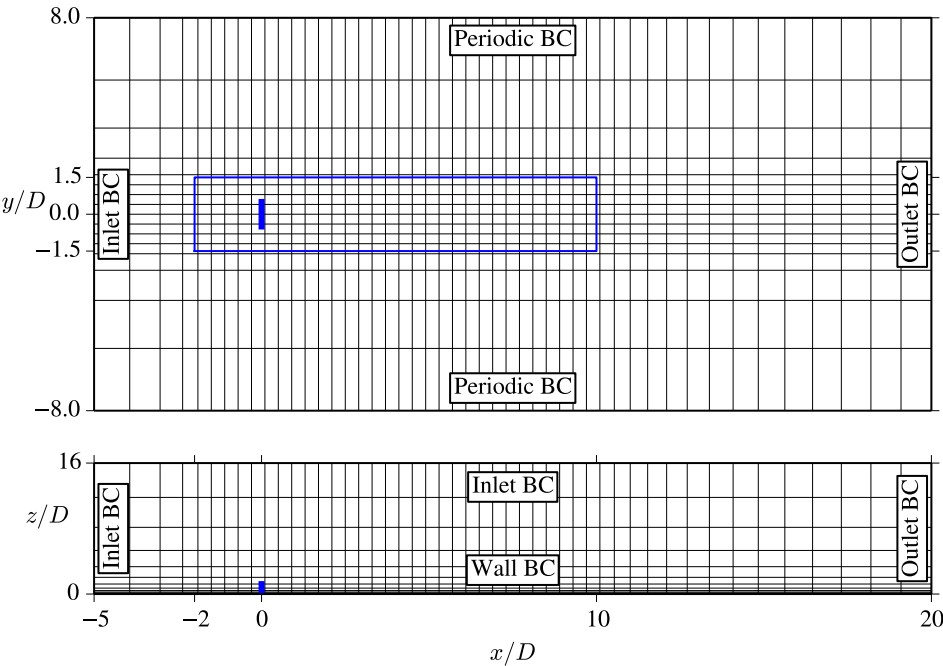

**Figure 3.** Grid and boundary conditions of the single AD simulation. Every 4th grid line is plotted. Top: top view of the grid. Bottom: side view of the grid. Blue filled box represents the AD. The horizontal spacing inside the blue rectangle is set to $D/8$.

## 3 Results and Discussion

### 3.1 Wind farm simulations in a neutral ABL

Figure 4 shows two contour plots of the stream-wise velocity, taken at hub height, for the two wind farm representations: 25 ADs and a high roughness. The $x$ and $y$ coordinates are normalized by the length or width of the wind farm $L_{WF} = 32D$.

5   The simulation with 25 ADs shows five distinct merged wakes, while the simulation of the roughness change shows one wind farm wake structure. A contour line that represents 95% recovered velocity is shown in each plot of Figure 4. The contour line reveals that the near wake of the wind farm represented by ADs is deflected clockwise, while the opposite is observed for the wind farm represented by a high roughness.

  In Figure 5, the turbulence intensity at hub height is plotted for both wind farm representations. The turbulence intensity is

10   larger and more concentrated for the wind farm represented by the 25 ADs compared to the wind farm represented by the high roughness. In addition, the increase in turbulence intensity at hub height is delayed in the roughness change simulation because the internal boundary layer (IBL) starts at $z = z_0$ (instead of $z = z_H$).

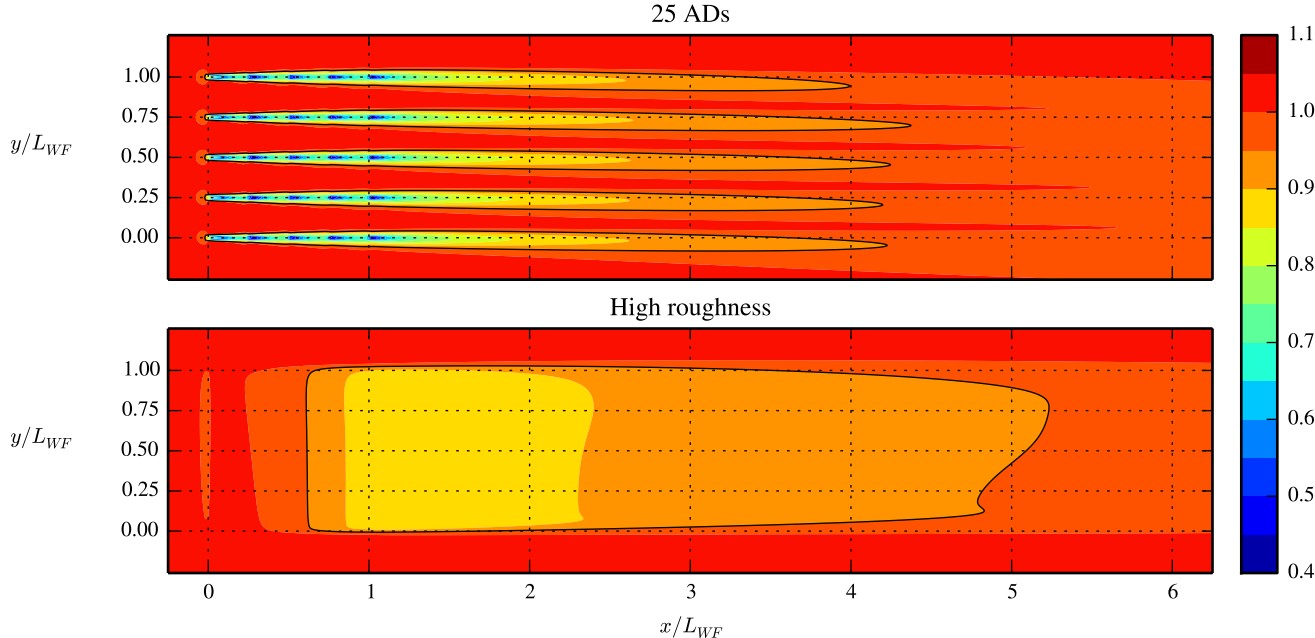

**Figure 4.** Stream-wise velocity at hub height, normalized by the free-stream. Top: Wind farm modeled with 25 ADs. Bottom: Wind farm modeled as a high roughness. Contour line represents 95% recovered velocity.

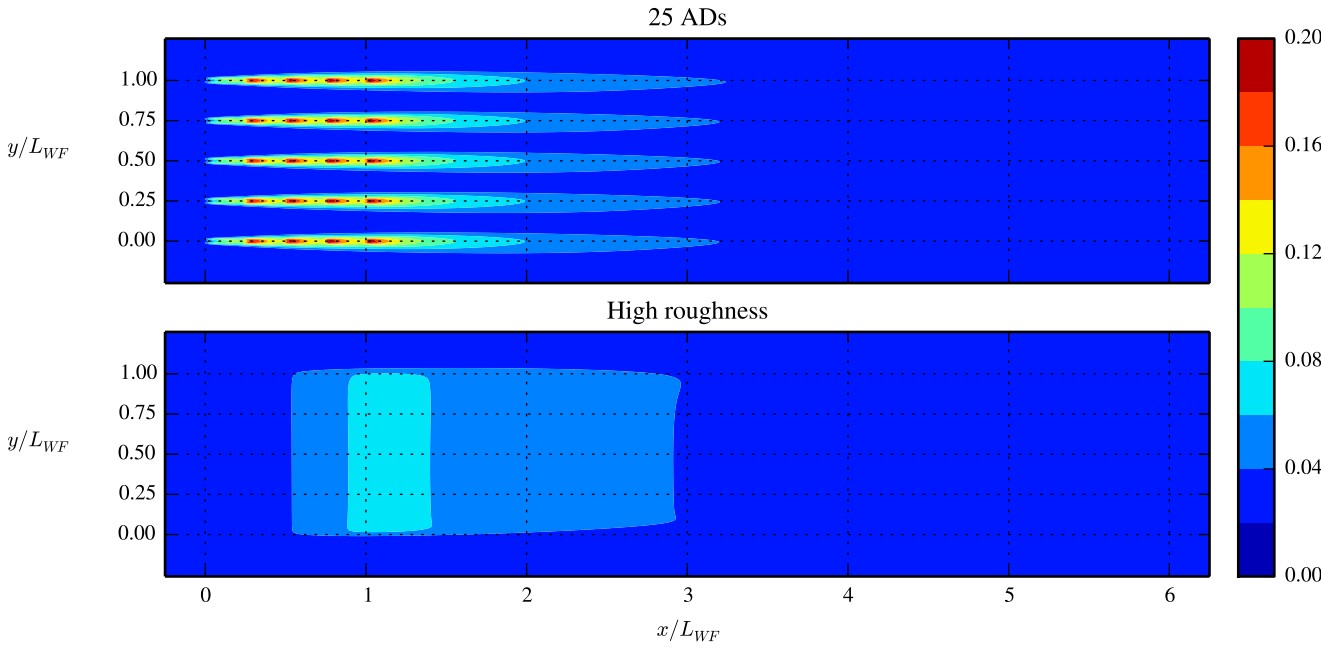

**Figure 5.** Turbulence intensity at hub height. Top: Wind farm modeled with 25 ADs. Bottom: Wind farm modeled as a high roughness.

The wind farm wake deflection is also visible in Figure 6, where contours of the stream-wise velocity, subtracted and normalized by the free-stream $(U - U_{\text{inflow}})/U_{\text{inflow}}$, are plotted at five cross planes located at $x/L_{WF} = 0, 0.1, 1, 2, 6$, for both simulations. Figure 6 shows that the far wake $(x/L_{WF} = 6)$ of the wind farm represented by a high roughness is turning back towards the free-stream, since an opposite roughness change occurs after the wind farm $(1 \text{ m} \rightarrow 10^{-4} \text{ m})$. The clockwise wake deflection of the wind farm represented by the 25 ADs is still visible at $x/L_{WF} = 6$.

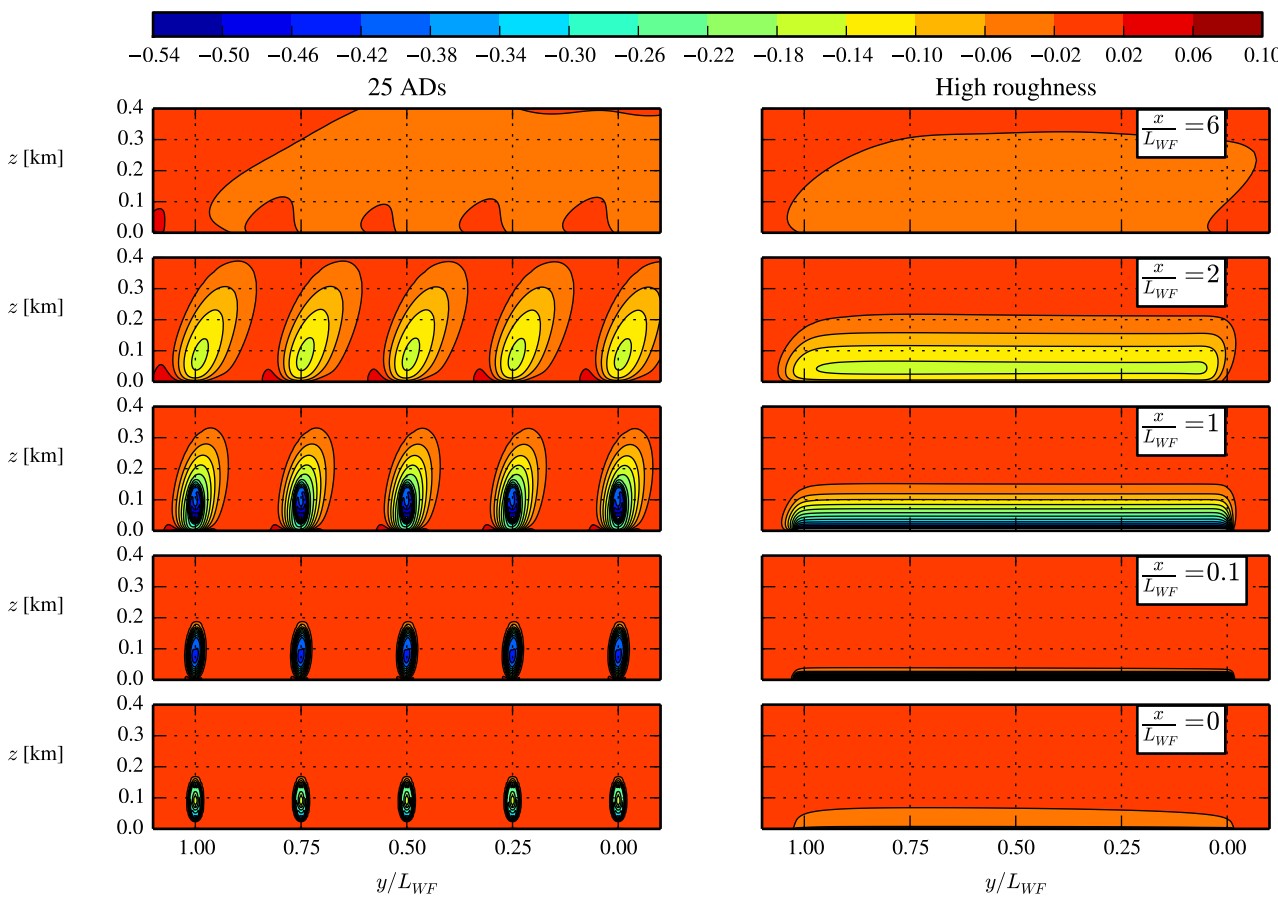

**Figure 6.** Stream-wise velocity subtracted and normalized by the free-stream $(U - U_{\text{inflow}})/U_{\text{inflow}}$ at several downstream cross planes. Left: wind farm is represented by 25 ADs. Right: wind farm is represented by a high roughness. Wind farm ends at $x/L_{WF} = 1$.

### 3.1.1 Momentum balance

In this section, the observed wind farm wake deflection from Figures 4 and 6 is explained using a momentum balance.

The momentum equation can be written as:

$$\frac{DU_i}{Dt} = S_{AD,i} + \frac{1}{\rho}\left(S_{v,i} - \frac{\partial \widetilde{P}}{\partial x_i}\right) - \frac{\partial \overline{u_i' u_j'}}{\partial x_j} \tag{3}$$

Here, we neglect the molecular viscosity since it is much smaller than the eddy viscosity. $S_{AD,i}$ represents the AD forces in the wind farm simulation including ADs, and $\widetilde{P}$ represents the fluctuation around the static pressure that is solved by the

5  SIMPLE algorithm from Patankar and Spalding (1972). The pressure gradients are obtained as $\partial P/\partial x = \partial \widetilde{P}/\partial x + \rho f_c V_G$ and $\partial P/\partial y = \partial \widetilde{P}/\partial y - \rho f_c U_G$. We are interested in the momentum balance in the cross direction $(y)$, which can be written as:

$$\underbrace{\int_A \frac{DV}{Dt}dA}_{\text{Imbalance of V-momentum}} = \underbrace{-f_c \int_A U dA}_{\text{Coriolis}} \quad \underbrace{-\frac{1}{\rho}\int_A \frac{\partial P}{\partial y}dA}_{\text{Pressure gradient}} \quad \underbrace{-\int_A \frac{\partial \overline{v'w'}}{\partial z}dA}_{\text{Turbulence}} \tag{4}$$

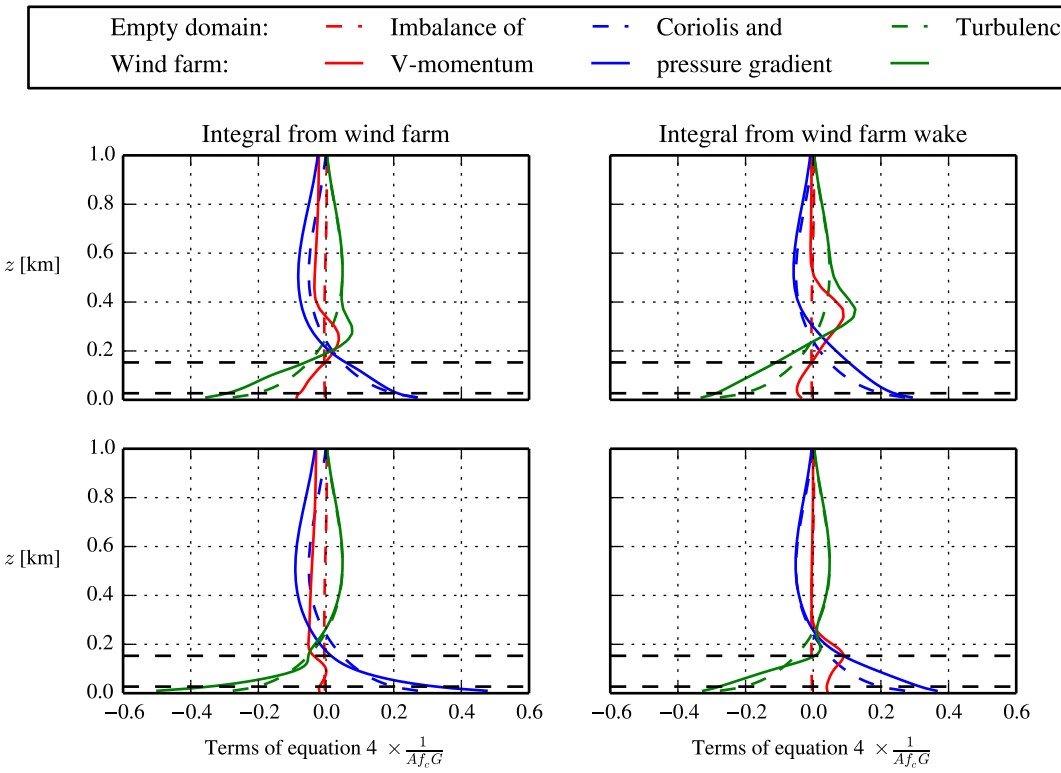

**Figure 7.** $V$-Momentum balance. Top: wind farm represented by 25 ADs, where rotor area is shown as black dashed lines. Bottom: wind farm represented by a high roughness. Left: integral taken from wind farm area. Right: integral taken from near wind farm wake. Dashed lines represent the results from an empty domain.

The integrals are taken over square horizontal slices with an area of $A = 40 \times 40D^2$ at several heights, in the wind farm $(x = y = \{-4D, L_{WF} + 4D\})$ and in the near wind farm wake $(x = \{L_{WF}, 2L_{WF} + 8D\}, y = \{-4D, L_{WF} + 4D\})$. The

integrals $\int_A \frac{\partial \overline{u'v'}}{\partial x} dA$ and $\int_A \frac{\partial \overline{v'v'}}{\partial y} dA$ are neglected because they are 2-3 orders of magnitude smaller than the other integrals from equation 4. In addition, $S_{AD,y} = 0$ since only thrust forces are considered, and the ADs are fixed normal to the flow direction. Each term of equation 4 is normalized by $1/(A f_c G)$ and plotted in Figure 7.

The top and bottom figures show results from the simulation where the wind farm is represented by 25 ADs and a high roughness, respectively. In addition, the results from the left figures are taken inside the wind farm, while the right figures are made in the near wind farm wake. The results from the wind farm simulation (solid lines) are compared with the results taken from an empty domain (colored dashed lines). When the wind farm is not present (colored dashed lines from Figure 7), the turbulence is in balance with the Coriolis force and pressure gradient, as expected. At the wind farm area (left plots of Figure 7), there is more turbulence developing when the wind farm is represented by ADs (left, top plot) compared to a wind farm represented by a roughness change (left, bottom plot). In the near wind farm wake (right plots of Figure 7), this difference in turbulence between the two wind farm simulations is even more pronounced.

When the wind farm is represented by a high roughness (bottom plots of Figure 7), an IBL develops from the abrupt roughness change at $x/L_{WF} = 0$. The largest changes in turbulence are mainly occurring near the wall due to IBL, where also the largest imbalance of $V$-momentum in the near wind farm wake (right, bottom plot) is observed. In addition, the near wake (right, bottom plot) shows that the combined change of Coriolis force and pressure gradient is larger than the change of the turbulence. Hence, the local changes in Coriolis force and pressure gradient deflect the wind farm wake anticlockwise, which is already visible in the wind farm area, as shown by the right plots of Figure 6.

When the wind farm is represented by 25 ADs (top plots of Figure 7), the turbulence and $V$-momentum change both near the wall and above the wind turbines. In both the wind farm and the near wind farm wake, the change in turbulence is larger than the combined change of Coriolis force and pressure gradient, especially above the wind farm, where also the imbalance of $V$-momentum in the near wind farm wake is the largest. This indicates that the turbulence is mixing flow from above the wind farm, that has a relative wind direction towards the right, down into the wake region, which causes the wind farm wake to turn clockwise. In other words, Figure 7 suggests that the Coriolis force is indirectly causing the wind farm wake to deflect clockwise because of the present wind veer, and not because of the local changes in the Coriolis force as motivated in previous work (van der Laan et al., 2015a).

Figure 7 shows that the flow in a simulation with 25 ADs including Coriolis is complex and very different from a simulation modeling a roughness change with Coriolis force. This means that the interaction between the Coriolis force and a wind farm wake cannot be simplified to the interaction between the Coriolis force and a roughness change, when the wake deflection is investigated, as suggested by Mitraszewski (2012).

### 3.1.2 Constant vs variable Coriolis forces

One could set the Coriolis force source terms to be constant in the horizontal directions (also at the wind farm) to isolate the effect of the wind veer on the wind farm wake deflection:

$$S_{v,x} = \rho f_c (V_{\text{precursor}} - V_G), \qquad S_{v,y} = -\rho f_c (U_{\text{precursor}} - U_G), \tag{5}$$

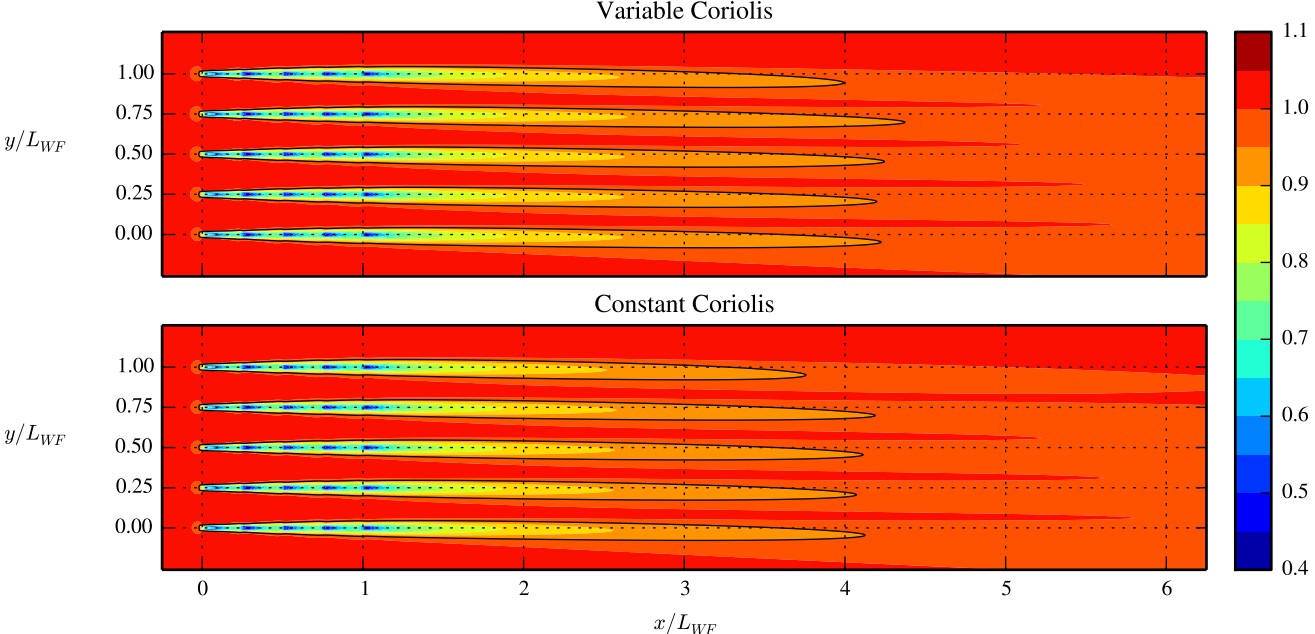

**Figure 8.** Stream-wise velocity at hub height, normalized by the free-stream. Wind farm represented by 25 ADs. Contour line represents 95% recovered velocity.

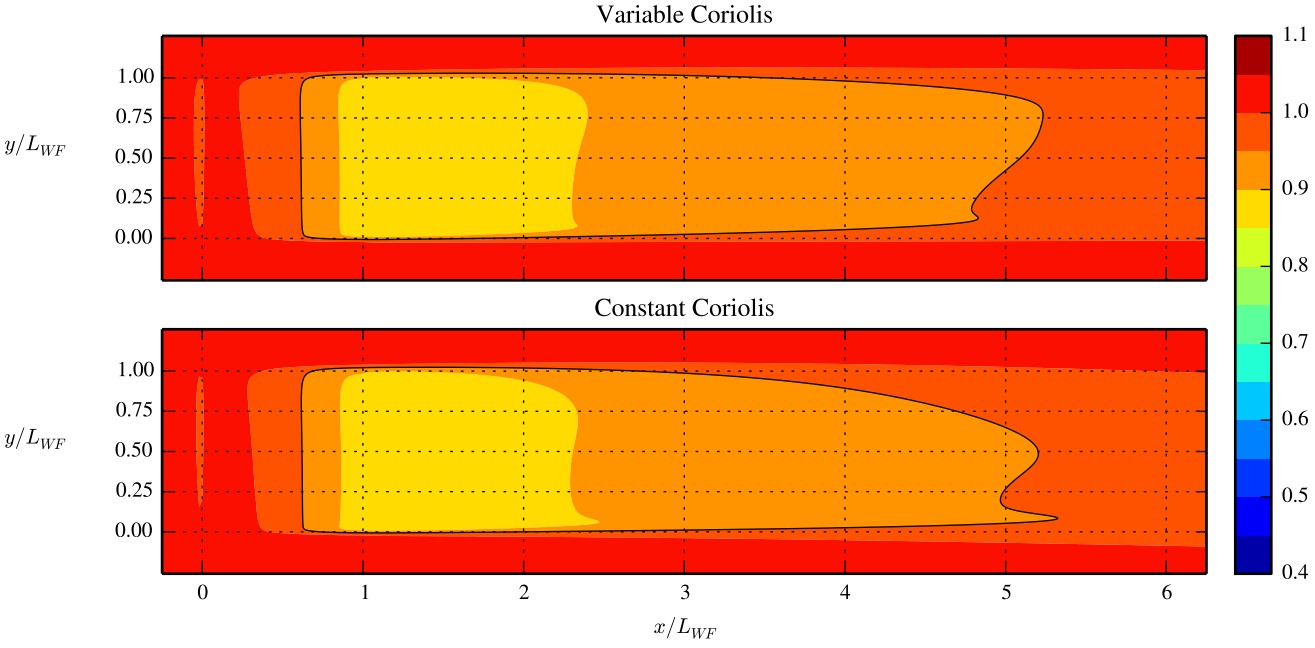

**Figure 9.** Stream-wise velocity at hub height, normalized by the free-stream. Wind farm represented by a high roughness. Contour line represents 95% recovered velocity.

where $U_{\text{precursor}}$ and $V_{\text{precursor}}$ are taken from the precursor simulation of the ABL profiles that are also used at the inlet boundary. (Without wind farm, the ABL profiles are maintained through out the domain.) The stream-wise velocity contours at hub height are plotted in Figures 8 and 9, for a wind farm represented by 25 ADs and a high roughness, respectively. In each figure, results with a variable (as already shown in Figure 4) and a constant Coriolis force (equation 5) are plotted. Figure 8 shows that there is hardly any visible difference in the wind farm wake deflection between a constant and a variable Coriolis force, which means that only the wind veer can be causing the clockwise wind farm wake deflection in a wind farm represented by 25 ADs. When the wind farm is modeled as a high roughness, as depicted in Figure 9, the difference between a constant and a variable Coriolis force is clearly visible. For a constant Coriolis force, the wake of the wind farm represented by the high roughness deflects more clockwise compared to the variable Coriolis force because a varying Coriolis force turns the wind farm wake anticlockwise, while the wind veer does the opposite. This shows that the locally changing Coriolis force is important in the simulation of the high roughness wind farm. Overall, Figure 8 is a proof that the wind veer is turning the wake of wind farm represented by 25 ADs clockwise, as previously concluded from the results of the momentum balance discussed in Section 3.1.1.

## 3.2 Single wake simulation in a shallow ABL

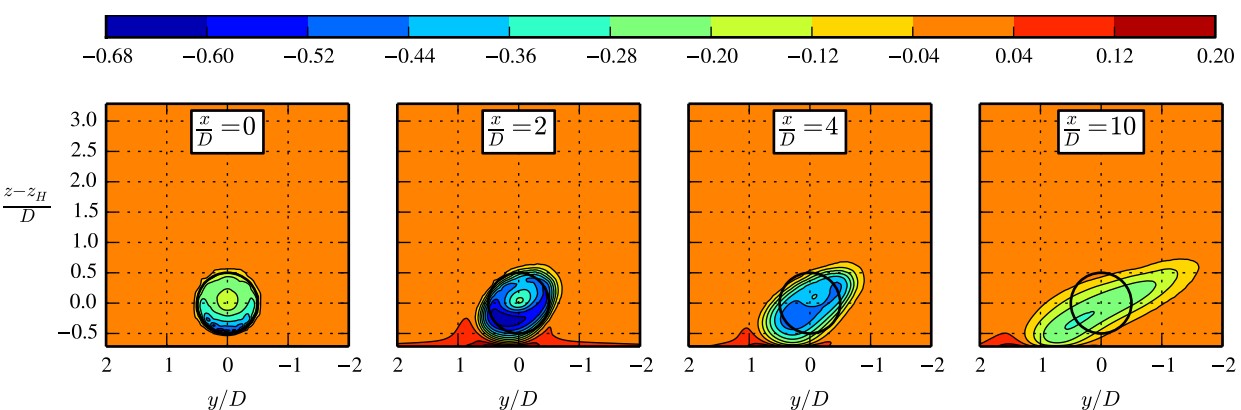

**Figure 10.** Stream-wise velocity subtracted and normalized by the free-stream $(U - U_{\text{inflow}})/U_{\text{inflow}}$ at several downstream cross planes for a single wake in a shallow ABL. AD is shown as a black circle.

Contours of stream-wise velocity, subtracted and normalized by the free-stream, of a single AD operating in a shallow ABL are shown in Figure 10, for 4 downstream cross planes located at $x/D = 0, 2, 4, 10$. Figure 10 shows that the strong wind veer stretches the wind turbine wake, which is observed as a clockwise rotation at hub height, as shown in Figure 11. A similar result has been observed in field measurements (Magnusson and Smedman, 1993) and LES of a single wind turbine and wind farms in a stable ABL (Lu and Porté-Agel, 2011; Churchfield et al., 2016; Abkar and Porté-Agel, 2016). The single AD test case confirms that the Coriolis force is indirectly deflecting a wind turbine/farm wake clockwise because of the wind veer. Note

that in neutral conditions, the wake deflection of single AD due to Coriolis is negligible because the wind veer is not strong enough, as shown in previous work (van der Laan et al., 2015a).

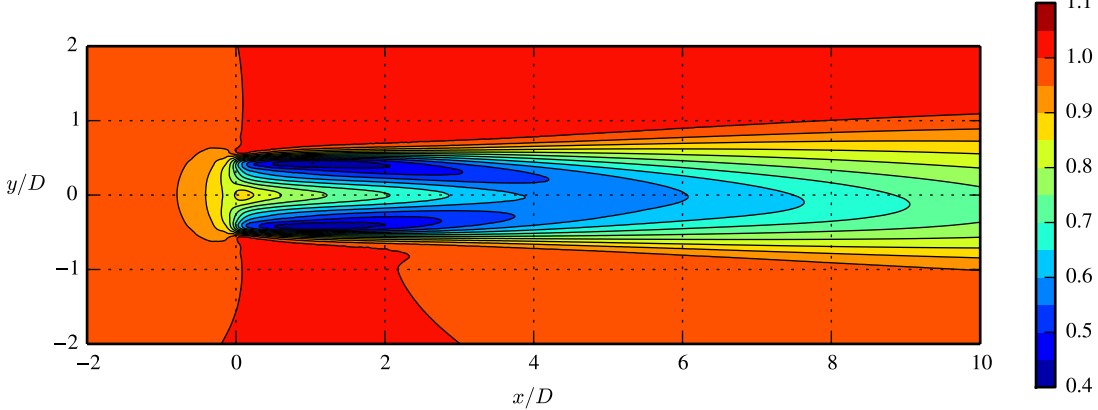

**Figure 11.** Stream-wise velocity at hub height, normalized by the free-stream, for a single wake in a shallow ABL.

## 4  Conclusions

Two RANS simulations of a wind farm including the effect of the Coriolis force are carried out, that differ in wind farm representation. When the wind farm is modeled as a roughness change, the wind farm wake turns anticlockwise due to an imbalance in the Coriolis force. When the wind farm is represented by 25 actuator disks, the wind farm wake is deflected clockwise. An investigation of the momentum balance in the cross flow direction suggests that in the simulation with 25 actuator disks, the turbulence is mixing momentum from above, that has a relative wind direction towards the right, down into the wake region. When the Coriolis force is set constant in the horizontal dimensions to isolate the effect of wind veer, the wind farm wake deflection the 25 actuator disks is unaffected. This proofs that the Coriolis force is indirectly causing the wind farm wake to deflect clockwise because of the present wind veer, and not because of the local changes in the Coriolis force, which is also confirmed by a simulation of a single actuator disk operating in a shallow atmospheric boundary layer. Hence, the interaction between the Coriolis force and a wind farm wake is a complex process that cannot be simplified to the interaction between the Coriolis force and a roughness change, when the deflection of the wind farm wake is investigated.

*Acknowledgements.* This work is supported by the Center for Computational Wind Turbine Aerodynamics and Atmospheric Turbulence funded by the Danish Council for Strategic Research, grant number 09-067216. Computational resources were provided by DCSC and the DTU central computing facility.

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
