# Peer review of "Why the Coriolis force turns a wind farm wake clockwise in the Northern Hemisphere"

_Wind Energy Science, 2016_

## Referee Comment (RC1) · Anonymous Referee #1 · 27 Feb 2017

Dear Author,

This is a very interesting article that point on a very interesting challenge in farm simulations.

I do agree with your conclusion but I would like to have a more extensive discussion on how one can make such a conclusion. In figure 6 you show very interesting results at two positions. I think it would be very interesting to see how the turbulence levels also vary with downstream position. You discuss that there are principal differences on how the farm is modeled and how the flow vary due to this. A plot of how the turbulence change with downstream position would ad a lot of understanding of what goes on. (Figure 4 with TI would be a fast solution)

On page 9, about line 7-11, you state that the Coriolis force is indirectly causing the wind farm wake deflect clockwise because the present wind veer. But I do believe you need to discuss how you can make such a conclusion more in depth. You also use the single wake example to verify your conclusion but it is not very clear how that supports your conclusion.

a minor comment: page 1, line 17, "curved rows" is not very clear.

In summary, I agree with the authors and think it is an interesting article that could be modified and accepted with limited effort. However, a deeper discussion on how these conclusions can be made is needed. The result in figure 6 in combination with the large differences in setup with individual turbines (disks) or roughness needs to supported by a discussion arguing that one can rule out other reasons. (Or at least say that they are of smaller order.)

Best regards

———————————————————

---

## Referee Comment (RC2) · Anonymous Referee #2 · 11 Mar 2017

General comments:

One of only a few papers for reviewing with not much to criticise... well done!

The paper addresses an important issue within the "wakes in the ABL" field and thus perfectly matches the scope of WES. The solution presented is consistent and reasonable. It provides a convincing approach for considering the Coriolis force and - more important - interpreting its impact on the wake flow. The paper's reasoning is clear and the modeling setup is appropriate for tackling the question. Based on the assumptions made, the workflow including the results is reproducable. As always when specific models are used it is hard to identify (as a reader) the influences due to model's life of its own.

[Figure]

Nevertheless, the paper is definitely a valuable contribution to the field.

Specific comments:

The roughness length for the wind farm has been chosen as 1m. Did you check for its influence on the analysis? E.g., by comparing with a smaller one?

Technically the paper is of good quality with no flaws in English language – only a couple of typos and small technical issues were found (see next paragraph).

Technical corrections:

p1, l21: deflections

p2, l6: as a result

p2, l29: boundary layer height

p3, l6: Blackadar

p3, l9: buoyancy

p4, l15: inserted as the inlet BC (?)

p5, l9: contour plots

p6, l4: Figure 5 shows

p6, l9: "...deflection from Figures 4 and 5 is explained ..." (no comma)

p7, eq.4: "Coriolis ... Pressure ... Turbulence and p8, l7: "turbulence is in balance with the Coriolis force and pressure gradient": imprecise naming: physically, it's a balance of accelerations

---

## Author Comment (AC1) · 16 Mar 2017

**Answer to Reviewer 1**

**March 16, 2017**

Thank you for reviewing the article. I have copied your comments in bold letters and answered them individually.

**I do agree with your conclusion but I would like to have a more extensive discussion on how one can make such a conclusion. In figure 6 you show very interesting results at two positions. I think it would be very interesting to see how the turbulence levels also vary with downstream position. You discuss that there are principal differences on how the farm is modeled and how the flow vary due to this. A plot of how the turbulence change with downstream position would ad a lot of understanding of what goes on. (Figure 4 with TI would be a fast solution)**

I have added two plots of turbulence intensity in Figures 1 and 2, which show that the levels of turbulence intensity in the simulation of the roughness change are much lower and smeared out compared to the turbulence intensity observed in the simulation including the 25 ADs. These plots confirm the results shown in Figure 6 of the submitted article. I think it is enough to add Figure 1 to the revised article.

**On page 9, about line 7-11, you state that the Coriolis force is indirectly causing the wind farm wake deflect clockwise because the present wind veer. But I do believe you need to discuss how you can make such a conclusion more in depth. You also use the single wake example to verify your conclusion but it is not very clear how that supports your conclusion.**

I can add the following (in italic font) to clarify the conclusion:

When the wind farm is represented by 25 ADs (top plots of Figure 6), the turbulence and $V$-momentum change both near the wall and above the wind turbines. In both the wind farm and the near wind farm wake, the change in turbulence is larger than the combined change of Coriolis force and pressure gradient, especially above the wind farm, where also the imbalance of $V$-momentum in the near wind farm wake is the largest. *This indicates that the turbulence is mixing flow from above the wind farm, down into the wake region. Since the flow above the wind farm has a relative wind direction towards the right due to the present wind veer, the wind farm is wake is turned clockwise.* In other words, Figure 6 suggests that the Coriolis force is indirectly causing the wind farm wake to deflect clockwise because of the present wind veer, and not because of the local changes in the Coriolis force as motivated in previous work (van der Laan et al., 2015a).

At moment, we have no other arguments that support this conclusion.
The single wake study shows that if the wind veer is strong, it turns even more the wake clockwise. This is an indication why the wind farm wake is also turning clockwise in the neutral case when the wind veer is less strong, but it is not a hard proof.

**a minor comment: page 1, line 17, "curved rows" is not very clear.**

The deflection of the upstream wind farm wake resulted in a lower power production of the downstream wind farm, because the Coriolis force aligned the upstream wind farm wake towards the curved *wind turbine* rows of the downstream wind farm.

**In summary, I agree with the authors and think it is an interesting article that could be modified and accepted with limited effort. However, a deeper discussion on how these conclusions can be made is needed. The result in figure 6 in combination with the large differences in setup with individual turbines (disks) or roughness needs to supported by a discussion arguing that one can rule out other reasons. (Or at least say that they are of smaller order.)**

I agree that our approach does not rule out the possibility of a wind farm wake turning the opposite direction in neutral conditions. Very recently, Allaerts and Meyers [1] have reported that a wind farm wake turns slightly counter clockwise (2°) in their large eddy simulations (LES) of a wind farm in neutral atmospheric conditions and argued that the local change in Coriolis force dominates the turbulent transport of spanwise momentum, rather than the wind veer. However, the observed turning is so small that it becomes challenging to extract it from a LES data set. This is because

the (neutral) atmospheric boundary layer that is inserted at the inlet is always developing downstream, which can lead to small wind direction changes in the wind farm. In our RANS simulations, the neutral ABL is in balance with the entire domain. In addition, one might need to simulate a very long time in LES in order to obtain a statistically independent average of a small quantify such as the wind direction deflection in neutral conditions. I think it is a good idea to add this discussion (and reference) in a revised version of the article.

[Figure]

Figure 1: Turbulence intensity at hub height. Top: Wind farm modeled with 25 ADs. Bottom: Wind farm modeled as a high roughness.

**References**

[1] Allaerts, D. and Meyers, J. Boundary-layer development and gravity waves in conventionally neutral wind farms. *Journal of Fluid Mechanics*, 814:95–130, 2017.

[Figure]

Figure 2: Turbulence intensity at several downstream cross planes. Left: wind farm is represented by 25 ADs. Right: wind farm is represented by a high roughness. Wind farm ends at $x/L_{WF} = 1$.

---

## Author Comment (AC2) · 16 Mar 2017

**Answer to Reviewer 2**

**March 16, 2017**

Thank you for reviewing the article. I have copied your main comments below in bold letters.

**Specific comments:**
**The roughness length for the wind farm has been chosen as 1m. Did you check for its influence on the analysis? E.g., by comparing with a smaller one? Technically the paper is of good quality with no flaws in English language only a couple of typos and small technical issues were found (see next paragraph).**

It is interesting to wonder what the roughness length of a wind farm should be and how it would influence the current study. Wind farm modelers have used values between 0.5-0.7 m [3, 4], using the wind farm roughness relation of Frandsen [2]. Studies based on large eddy simulations of a fully developed boundary layer over an infinitely large wind farm have shown that the wind farm roughness can be as high as 8.9 m [1]. Hence, using 1 m as a wind farm roughness in the present work seems to be in the correct range. Choosing a different roughness will change the results, but not the general trends observed in the simulation of wind farm represent by a roughness length. I have performed simulations of wind farm roughness of 10 m, which gave a more stronger initial counter clockwise deflection compared to using a wind farm roughness of 1 m. I will add a discussion about the wind farm roughness in the revised article. In addition, I will correct the article for the typos that you have pointed out.

**References**

[1] Calaf, M., Meneveau, C., and Meyers, J. Large eddy simulation study of fully developed wind-turbine array boundary layers. *Physics of Fluids*, 22:015110, 2010.

[2] Frandsen, S. T. Turbulence and turbulence-generated structural loading in wind turbine clusters. Risø-R-1188(en), Risø National Laboratory, 2007.

[3] Frandsen, S. T., Jørgensen, H. E., Barthelmie, R., Rathmann, O., Badger, J., Hansen, K., Ott, S., Réthoré, P.-E., Larsen, S. E., and Jensen, L. E. The Making of a Second-generation Wind Farm Efficiency Model Complex. *Wind Energy*, 12:445–458, 2009.

[4] Volker, P. *Wake Effects of Large Offshore Wind Farms - a study of the Mesoscale Atmosphere*. PhD thesis, DTU Wind Energy, 2014.

---

## Author Comment (AC3) · 17 Mar 2017

**Follow up on answer to Reviewer 1**

March 17, 2017

In the previous answer to Reviewer 1, I have mentioned that we have no additional arguments that support our conclusion of the influence of the wind veer on the wind farm wake deflection. I would like to follow up on this by performing an additional study that rules out the influence of the local changing Coriolis force on the wake deflection due to the wake deficits. One could set the Coriolis force source terms to be constant in the horizontal directions (also at the wind farm) as an experiment:

$$S_{v,x} = \rho f_c \left( V_{\text{precursor}} - V_G \right), \qquad S_{v,y} = -\rho f_c \left( U_{\text{precursor}} - U_G \right), \tag{1}$$

where $U_{\text{precursor}}$ and $V_{\text{precursor}}$ are taken from the precursor simulation of the ABL profiles that are also used at the inlet boundary. (Without wind farm, the ABL profiles are maintained through out the domain.) The stream-wise velocity contours at hub height are plotted in Figures 1 and 2, for a wind farm represented by 25 ADs and a high roughness, respectively. In each figure, results with a variable (as used in the submitted article) and a constant Coriolis force (equation 1) are shown. Figure 1 show that there is hardly any difference between a constant and a variable Coriolis force, which means that only the wind veer can be causing the clockwise wind farm wake deflection. When the wind farm is modeled as a high roughness, as depicted in Figure 2, the difference between a constant and a variable Coriolis force is clearly visible. For a constant Coriolis force, the wake of the wind farm represented by the high roughness deflects more clockwise compared to the constant Coriolis case because a varying Coriolis force turns the wind farm wake counter clockwise, while the wind veer does the opposite. This shows that the locally changing Coriolis force is important in the simulation of the high roughness wind farm. I think it would be useful to include this extra study in the article because it is a proof that the interaction of the wind farm wake and the wind veer is the main mechanism why the wind farm wake deflects clockwise.

[Figure]

Figure 1: Stream-wise velocity at hub height, normalized by the free-stream. Wind farm modeled with 25 ADs.

[Figure]

Figure 2: Stream-wise velocity at hub height, normalized by the free-stream. Wind farm modeled with a high roughness.